# OpenReview forum: "AgentMonitor: A Plug-and-Play Framework for Predictive and Secure Multi-Agent Systems"
_ICLR.cc/2025/Conference — ICLR 2025 Conference Withdrawn Submission_

### Official Review · Reviewer_NFRZ · 2024-10-23

**Soundness:** 2
**Presentation:** 2
**Contribution:** 2
**Rating:** 5
**Confidence:** 3

**Summary:**

This paper proposes a new method for predicting the performance of an MAS with a specific architecture on a given task. To this end, the authors introduce the AgentMonitor, which takes the inputs and outputs of all agents and computes different indicators of this MAS. Then, these indicators along with the target score form a data point, which is used to train an XGBoost model. This XGBoost model will then be used to predict the target score of an unseen MAS on the given task. Besides, as AgentMonitor can receive the inputs and outputs of each agent, it can be used to correct the outputs of each agent via post-editing, which can improve the harmlessness of the MAS when malicious agents exist. Experiments show the effectiveness of AgentMonitor as a plug-and-play framework in predicting the performance of an MAS and improving the security of the MAS.

**Strengths:**

The idea is intuitive. The problem of accurately predicting the performance of an MAS with a specific architecture on a given task is clear. Another advantage of AgentMonitor is the ability to defend against malicious agents in a MAS via editing the inputs/outputs of agents.

**Weaknesses:**

The presentation and the overall flow of the paper are somewhat messy, making it hard to grasp and fully understand the details, especially the evaluation process. Following are some questions, and please correct me if I make mistakes.

1. Are you using the same LLM for both the agent and the judger (gives the values of the indicators)? Or they can be different? If so, how will the choice of LLM influence the performance of the MAS?

2. There are many indicators used in experiments. Although there is an ablation study on the top 8 and bottom 8 indicators, it is still unclear why using so many indicators is necessary and how will these different indicators influence the outcomes.

3. In Section 3.3, I am confused by the RQ2. In the training set, each data point is (indicators, performance), where the indicators are obtained by feeding the full problem-solving process to the judger (an LLM). Then, what does "instances" mean? I assume that the prediction process is as follows: Each task-solving in the \textbf{test set} corresponds to a data point (indicators, ground-truth performance) where the indicators are obtained using the same method as in the training set. Then, we feed the indicators to the XGBoost model to get the predicted performance. Then, we can compare the predicted performance with the ground-truth performance. Therefore, why do we need to use a set of "instances" to calculate the indicators (which are obtained by the judger)?

4. For Section 3.2, I can hardly understand Figure 8. What do the numbers in the figure mean? What is the meaning of the axes? The explanations for the results are weird and hard to understand. "the indicators with higher importance have a wider range of valid choices", what does "valid choice" mean?

5. The paper needs substantial revision in terms of the presentation. A better organization is required to more clearly show the details and experimental results. I suggest more formally presenting the method part, which would greatly improve the readability of the paper.

Other minors:

Figure 1 is not referenced throughout the paper.

All the figures are hard to read; the text is too small to grasp.

**Questions:**

See Weaknesses.

---

### Official Review · Reviewer_WP9M · 2024-11-02

**Soundness:** 3
**Presentation:** 3
**Contribution:** 2
**Rating:** 5
**Confidence:** 4

**Summary:**

This paper presents AgentMonitor, a framework that integrates with multi-agent systems (MAS) to provide predictive monitoring and enhance security through real-time post-editing. The tool captures agent-level and graph-level indicators to build a regression model that predicts task performance, aiming to reduce trial-and-error in configuring MAS. AgentMonitor also features a mechanism for on-the-fly correction of agent outputs to mitigate harmful or unhelpful results. The paper demonstrates empirical results across several MAS tasks, including coding, language understanding, and mathematical problem-solving, showcasing improvements in system reliability and safety.

**Strengths:**

1. The integration of performance prediction with a post-editing layer for real-time response correction is novel and adds practical value to MAS.

2. The system's ability to integrate without altering the existing MAS infrastructure makes it a versatile solution for developers.

3. The framework is tested on various MAS architectures and demonstrates resilience, improving performance metrics like helpfulness and reducing harmful outputs by significant margins.

**Weaknesses:**

1. While the combination of predictive monitoring and post-editing is valuable, the paper could better articulate what sets AgentMonitor apart from existing MAS monitoring tools and frameworks.


2. The AgentMonitor framework was trained and tested on the same or similar task types. While the experiments demonstrate strong performance in specific MAS tasks (e.g., coding challenges and problem-solving tests), this approach raises concerns about overfitting and a limited ability to generalize to more diverse and complex tasks. The framework's predictive accuracy and real-time correction mechanisms may be tailored too closely to the tested scenarios, making it unclear how AgentMonitor would perform in more varied or practical real-world MAS deployments.

**Questions:**

1. What are the specific novel aspects of the AgentMonitor framework compared to existing monitoring and predictive tools for MAS? How does it improve upon prior approaches in terms of performance prediction and on-the-fly corrections?

2. Given that the framework was trained and tested on the same or similar types of tasks, how confident are you in AgentMonitor’s performance when applied to different or more complex MAS tasks outside the scope of your experiments?

3. How were the specific agent-level and graph-level indicators chosen and how they impact the predictive model’s accuracy?

---

### Official Review · Reviewer_5t2e · 2024-11-02

**Soundness:** 2
**Presentation:** 2
**Contribution:** 2
**Rating:** 3
**Confidence:** 3

**Summary:**

The paper presents AgentMonitor, a framework that aims to improve the predictability and security of multi-agent systems (MAS). It is designed to be "plug-and-play" meaning it can be integrated with existing multi-agent applications.

**Strengths:**

- The framework’s plug-and-play design allows it to integrate easily with existing MAS setups without significant modifications.
- By using indicators to predict MAS performance, AgentMonitor offers a predictive solution, reducing the need for trial-and-error configurations.

**Weaknesses:**

- The section 2.1, Design of AgentMonitor, lacks specific design information, making it difficult to understand the authors' intentions. The justification for using LoRA appears unnecessary, and merely referencing the appendix for details on AgentMonitor’s usage does not clarify its design. Key elements, such as how agent input/output is captured and the rationale behind design choices, are missing.
- In section 2.3, Post-Edit Features, the motivation and design of the post-edit feature are not clearly explained. The reference to Figure 4 lacks meaningful context, and simply stating how AgentMonitor turns “BAD” into “GOOD” is insufficient. The authors should provide more in-depth information on how the framework effectively manages and controls undesirable agent behavior.
- The approach's effectiveness in larger, more complex MAS configurations isn’t fully explored. Further testing in more extensive/complex MAS environments would clarify its scalability.
- Many crucial details on indicators and the experimental setup are presented in the appendix; these should be included in the main paper.

**Questions:**

1. Can you provide more details on the actual internal design of AgentMonitor?
2. Can AgentMonitor handle a significantly larger number of agents without sacrificing prediction accuracy or real-time correction capabilities?
3. Have you tested the robustness of AgentMonitor against out-of-domain or noisy data? For instance, have you tried synthetically generating these types of data to evaluate how well the system maintains prediction accuracy and security in non-ideal conditions?
4. Have you considered adding more real-world datasets to further test the framework? I suspect that as the data becomes more diverse, the correlation might decrease, potentially revealing areas for model refinement or additional indicator development.

---

### Official Review · Reviewer_uhes · 2024-11-02

**Soundness:** 2
**Presentation:** 3
**Contribution:** 2
**Rating:** 5
**Confidence:** 4

**Summary:**

The paper proposes and studies the problem of predicting the performance of MAS from (a) features that encode information about the architectture of MAS (b) performance of individual systems within MAS. They propose an approach in which relatively simple features of MAS are identified and fed to XGboost based regressor which is trained to predict performacne based on the values of input features.

**Strengths:**

- The problem studied by this work is novel and interesting.
- The figures are nicely done.
- The writing is generally ok.

**Weaknesses:**

- The metric that the paper is using primarily is pearson correlation -- this choice of metric is confusing to me. If the point is to understand how well AgentMonitor predicts the relative ranking of systems, perhaps Kendall Tau correlation would be a better choice (https://en.wikipedia.org/wiki/Kendall_tau_rank_correlation_coefficient).

- I am generally dissatisfied by the evaluation and how the results are presented. For example, figure 8 is a novel plot type (i.e., not common in ML papers) but no information is provided on how the plot should be interepreted, 8a is a very messy plot anyways. May be the readers are not actually meant to understand the plots, but only observe the density of lines within them, but this is not clear upfront.

- I am also confused about why authors think its interesting to look at correlation between features/indicators. It is natural and reasonable to expect that some features would be correlated; some would not be, and findings don't contradict that. I would have instead liked to see a more elaborate discussion of feature importance across different scenarios.

- The generalization of the method across architectures and tasks -- which is what we would want ideally from such a method is quite weak.

- The problem that the paper is trying to address is not made clear in the paper -- seems like authors have two applications in mind (a) monitoring at test time (based on the fact that it is called AgentMonitor) (b) predicting performance of novel MAS (based on discussion in related works section). Both are nice applications, but I don't think there is enough evidence provided to argue that this is the right approach for either application.

**Questions:**

1. Looking at table 2, all architecture seems to have one coder agent and figure 8a indicates it is the most important feature as well. To what extent the performance of AgentMonitor is recovered, if we only look at the information relevant to the primary coder agent?

2. it should be the error is not **high** (line 321)?
> Furthermore, although the correlation for HumanEval in In-Task setting is lower, the error is not,

3. How is the post-edit done in section 3.4? My understanding is that it is an 'independent' agent that takes the final response and edits it? If so, this can be done with any architecture with or without AgentMonitor, right?

---

### Official Review · Reviewer_obEf · 2024-11-06

**Soundness:** 3
**Presentation:** 3
**Contribution:** 3
**Rating:** 6
**Confidence:** 4

**Summary:**

The authors propose AgentMonitor, an input/output monitoring and transformation drop-in component that can be retrofitted to MAS. AgentMonitor can both help with systems configuration, as well as with defending against outputs from certain classes of malicious actors. On a series of toy benchmark environments, including coding and multi-agent reasoning tasks, the authors show that their AgentMonitor achieves some degree of generalisation across tasks, as well as can improve systems performance in the presence of malicious actors.

**Strengths:**

The authors suggest a practical approach that can be retrofitted to existing MAS in order to address to urgent challenges in practical MAS systems use and design:

* systems configuration
* malicious actors

The authors' suggested solution is simple to implement, and the empirical evaluation is promising.

**Weaknesses:**

* The simplicity of the authors' approach is also in itself a weakness: there isn't much technical novelty going on; systems monitoring with LLMs and input/output filtering are standard approaches, although perhaps not much work has investigated these in multi-agent systems.

* as not many multi-agent LLM systems have been deployed yet, the empirical results presented by the authors are naturally of limited generalisability to real-world tasks of importance; therefore while the authors' approach is promising in these limited settings, it remains unclear how much their simple approach would unlock in more impactful settings, particularly cf. more intensive approaches involving post-training etc.

**Questions:**

* A central issue with monitoring systems is the performance overhead / cost / latency that they incur. Could the authors on this aspect of their proposed solution?

---

### Note · Authors · 2024-11-25

**Comment:**

Thank you for the valuable suggestions provided by all reviewers. I plan to enhance our work by: 1) analyzing our framework in a more diverse and complex scenario; 2) reorganizing the paper to address certain issues in the methods section; and 3) providing a more in-depth analysis of the evaluation results. Hope that we can have a improved version in the future.

**Withdrawal Confirmation:**

I have read and agree with the venue's withdrawal policy on behalf of myself and my co-authors.